# Maladaptive Personality Traits in a Group of Patients with Substance Use Disorder and ADHD

**DOI:** 10.3390/medicina58070962

**Published:** 2022-07-20

**Authors:** Alexandra Mariana Buică, Diana Monica Preda, Lucia Emanuela Andrei, Mihaela Stancu, Nicolae Gică, Florina Rad

**Affiliations:** 1Child and Adolescent Psychiatry Department, “Carol Davila” University of Medicine and Pharmacy, 020021 Bucharest, Romania; alexa_de_du@yahoo.com (A.M.B.); ms.stancu@yahoo.ro (M.S.); florina2rad@yahoo.com (F.R.); 2“Grigore Alexandrescu” Emergency Children Hospital, 10556 Bucharest, Romania; diana_monica_preda@yahoo.com; 3Obstetrics and Gynecology Department, University of Medicine and Pharmacy “Carol Davila”, 020021 Bucharest, Romania; gica.nicolae@gmail.com

**Keywords:** Substance Use Disorder (SUD), early adulthood, comorbidity

## Abstract

*Background and Objectives*: the comorbidity of personality disorders in patients who use psychoactive substances is common in psychiatric practice. The epidemiology of disharmonious personality traits in patients with ADHD and addictions in adulthood is still insufficiently researched. The study investigated the typology of personality traits in a group of adult patients consuming psychoactive substances, in whom symptoms of ADHD were identified. *Materials and Methods*: the study evaluates a group of 104 patients with chronic psychoactive substances abuse, in whom symptoms of ADHD were identified in early adulthood, in terms of comorbid personality traits. *Results*: statistically significant data have been obtained regarding the presence of clinical traits characteristic for cluster B personality disorders, the patients presenting lower levels of self-control, self-image instability, difficulties in the areas of social relationships and own identity integration. *Conclusions*: ADHD symptomatology precedes the clinical traits of personality disorders in patients with addictions, negatively influencing chronic evolution and quality of life.

## 1. Introduction

Substance Use Disorder (SUD) is a psychiatric condition, which involves a modification of cerebral physiology through epigenetic mechanisms, during the transition from occasional, recreational consumption to repetitive, addictive behaviours of psychoactive substances [1]. Multiple epigenetic mechanisms that determine an aberrant neuroplasticity secondary to addictive behaviours are reported, the common characteristic of all drugs being their action on the dopaminergic circuit, known as the reward system [2].

During the past few years, numerous studies on the comorbidity of substance abuse and ADHD (Attention Deficit Hyperactivity Disorder) have been published, however, research on the relationship between substance use, ADHD and personality disorders continues to be poorly developed.

Studies that have investigated the co-occurrence of these disorders, report comorbidity rates ranging from 25% to 55% in adults with ADHD who had a history of psychostimulant abuse/dependence [3]. If we consider the prevalence rate in the general population (15–18%), we can deduce that the presence of ADHD in the general population increases twice the vulnerability to develop psychoactive substances abuse [4]. These patients are more likely to start substance abuse early in life, to simultaneously consume more drugs and to require longer periods of time in specific pharmacological and psychotherapeutic programmes [5]. Additionally, as reported by previous studies, patients who start using substances earlier in life are at risk for developing serious secondary psychiatric disorders, such as psychoses. Two studies from 2021, by Ricci et al. reported a link between prolonged cannabis use and development of first-episode psychosis associated with higher degrees of dissociative symptoms, observing higher DUP (Duration of Untreated Psychosis) in these patients as well [6,7].

Addiction negatively affects the evolution of ADHD with age, through complications such as cognitive deterioration and memory problems. On the other hand, substance abuse can mimic the symptoms of ADHD and in this way lead to overdiagnosis. The objective is to establish a set of ADHD symptoms, in addition to the symptoms related to the onset, nature, and severity of substance abuse, so that patients with comorbidities can receive adequate treatment. Addiction in patients with ADHD has a more severe and chronic course compared to those who have addictions but do not associate this disorder as well [8].

The presence of ADHD and personality disorder comorbidity, in patients with psychoactive substances abuse can be explained by the common impulsivity factor [9]. The overlap of genetic and environmental aetiology represents the common ground for the mechanisms involved in impulsive behaviours in the two diagnostic entities; in the case of consumers these are characterized by the inability to stop or control the repeated consumption of psychostimulating substances despite the unfavourable social, professional, or medical consequences, while in the case of ADHD patients’ manifest deficits in executive control [10].

Due to the fact that the diagnosis of ADHD in adults is a relatively new one, the overlapping impulsivity, mood instability, and aggressivity symptoms can easily lead to the diagnosis of personality disorder without the suspicion of ADHD being taken into account [11]. The investigation of these comorbidities brings advantages in choosing adequate psychotherapeutical and pharmacological intervention [12].

Certain personality traits based on impulsivity have been reported in multiple studies that have investigated patients with ADHD and SUD, such as the deficient self-control abilities [13], repeated sensation-seeking behaviours [14], the tendency to act according to one’s first impulse without taking possible consequences into account [15]. These results underline the presence of personality traits in patients with these comorbidities, but further research is still needed to assess how impulsive personality traits influence ADHD symptoms and the use of psychoactive substances.

Personality disorders have a prevalence reported in the literature between 10 and 15% in the general population [16], the diagnosis being found in up to 50% of clinical psychiatric patients [17]. The clinical picture develops in adolescence, the early onset of symptoms being a predisposing factor to risky behaviours later in life, in spite of the tendency being to improve with age. Pathological personality traits can alter these patients’ course and cause dysfunctionality in various areas of life even if, with age, the symptoms may manifest at subclinical intensity [18]. Certain personality features are considered, by some authors, the underlying mechanism in patients with dual diagnosis of psychotic disorders and SUD. In a 2021 review by Oh et al., after a meta-analysis on personality traits of patients with psychotic disorders and SUD comorbidity (dual diagnosis-DD), the authors found that patients with DD presented “elevated negative urgency”, “low premeditation”, as well as “elevated unconscientious disinhibition” [19].

The main objective of this study is to determine whether there is a significant relationship between personality disorders and drug use/ADHD. The secondary objective was to establish the most common type of personality disorder involved in this relationship.

## 2. Materials and Methods

Our current research is a continuation of a previously published study, where we highlighted that in a group of 104 patients with SUDs, that 46% of them met the criteria for adult ADHD [17].

### 2.1. Participants

The same group of 104 patients was included in this study. The patients, aged between 18–28 years, were selected from an addiction psychiatry unit at “Prof. Dr. Alexandru Obregia” Clinical Psychiatry Hospital in Bucharest, Romania. The age of the adult patients included in the study is between 18–25 years. 75% of the subjects were between 18 and 22 years old, with 21 years old being the age with most representatives. This age range was selected as the ward within which we conducted our research is dedicated to younger adults. Moreover, the majority of those who struggle with these disorders and who are referred to the aforementioned addictions clinic are aged 18–25 and they were generally more inclined to cooperate and consent to the inclusion in the study. The selection protocol of the subjects included in the study and research methodology were applied with the agreement of the Ethics Committee within “Carol Davila” University of Medicine and Pharmacy and “Prof. Dr. Alexandru Obregia” Clinical Psychiatry Hospital. In this cross-sectional, observational study we wanted to evaluate the pathological personality processes that were associated with the symptomatic picture of ADHD, and substance abuse rates.

Thus, the subjects were divided into two groups:-Control group: patients with SUD and no ADHD symptoms, consisting of 60 participants.-Study group: patients with SUD and ADHD symptoms, consisting of 44 participants.

All the participants underwent a comprehensive psychiatric assessment performed by a psychiatrist, consisting of the medical and developmental history, the educational history, the mental status examination, a Diagnostic and Statistical Manual of Mental Disorders, Fifth Edition (DSM-5) diagnosis and the treatment history.

All patients from this study used drugs chronically.

### 2.2. Procedures and Instruments

The evaluation of ADHD symptoms was carried out with DIVA 2.0, a structured interview approved in Romania, that investigates ADHD symptoms in adults, conceptualized in 2010 by J.J.S. Kooij [20]. This interview is based on the DSM criteria for ADHD and evaluates 18 criteria for current and retrospective (from childhood) behaviours. Deficiencies in the areas of functioning of daily life are also assessed. Testing is recommended in the presence of a family member to simultaneously evaluate heteroanamnestic information. The 3-part questionnaire assesses attention deficit, hyperactivity/impulsivity and age of onset/dysfunction associated with symptoms. Responses are adjusted to recent time (presence of symptom in the last 6 months) and retrospectively (presence of symptom between 5 and 12 years). For each symptom, its presence or absence is established in both stages of life. For scoring, it is established whether for each of the 3 parts at least 6 criteria have been registered. It is also recorded if there is evidence of continuity of the symptom in life, if it is associated with dysfunctions, if the dysfunctions are present in at least 2 areas of life and if the symptoms could be better explained by another psychiatric entity. The results regarding the percentual distribution of ADHD symptoms, the type of psychoactive substance depending on the predominant ADHD type, and ADHD evolution into adulthood were previously published [21].

The evaluation of maladaptive personality traits was performed applying Severity Indices of Personality Problems (SIPP-SF), developed in 2008 by Verheul and Collaborators [22]. This self-report instrument investigates the core components of (mal)adaptive personality functioning. The short version of the questionnaire was used, which contains 60 items divided into five distinct domains: self-control, identity, responsibility, relational capacities, social concordance. This self-evaluation questionnaire includes items represented by statements regarding the personality of the subject, the way of perception of life and of oneself. For each item, the subjects were asked to select to what extent the respective statement had applied to them in the last 3 months (total disagreement, partial disagreement, partial agreement, total agreement).

Statistical analysis: compiling the variables database was carried out with Microsoft Office Excel 2007. Statistical data were obtained using IBM SPSS Statistics 20, by performing descriptive and inferential statistical methods. We used histograms for graphical representation of the discrete quantitative variables. Descriptive analysis of the quantitative variables was performed, and the relationship between 2 quantitative variables was detailed using Pearson correlation coefficient and the relationship between a quantitative and a qualitative variable was tested with *t*-test for independent samples. For comparison of proportions, we used the z test, with a *p* < 0.05 significance level.

## 3. Results

The scores obtained for the items of personality traits questionnaire were analysed according to the presence or absence of diagnostic criteria for ADHD. The values for the Self-control section in the ADHD group were between 14 and 46, with an average+/−SD = 29.44+/−7.35, while in the group of those without ADHD they were between 17 and 48, with an average+/−SD = 33.44+/−8.58. A *t*-test for independent samples showed that there was a statistically significant difference between the average of the scores for the self-control field in those without ADHD and those with ADHD (t = 2.56, *p* = 0.01) (Figure 1).

There was no statistically significant difference between the two groups regarding the scores obtained for Identity integrity and Responsibility sections.

The score values for the Relational Skills field in those with ADHD were between 18 and 47, with an average+/−SD = 33.76+/−7.68 and in those without ADHD were between 26 and 47, with an average+/−SD = 36.96+/−6.07. A *t*-test for independent samples showed that there was a statistically significant difference between the 2 averages (t = 2.34, *p* = 0.02) (Figure 2).

The score values for the Social Adequacy domain in subjects with ADHD were between 19 and 48, with an average+/−SD = 34.41+/−6.89 while in subjects without ADHD were between 20 and 48, with an average+/−SD = 36.92+/−8.47. A *t*-test for independent samples showed that there was no statistically significant difference between the 2 averages (t = 1.66, *p* = 0.09) (Figure 3).

In Table 1, the Pearson correlation coefficient and the *p*-value are represented. There was a linear, positive, moderate, statistically significant correlation between the values obtained by the subjects for the Self-control section and those for the Identity Integration and Relational Skills domains. There was also a linear, positive, strong, statistically significant correlation between the values obtained by the subjects for the field of Self-control and the areas of Responsibility and Social Adequacy. There was a linear, positive, moderate, statistically significant correlation between the values for the Identity Integration section and those for the Responsibility domain. There was also a linear, positive, strong, statistically significant correlation between the values of the Identity Integration domain and those of the Relational Skills and Social Adequacy domains.

## 4. Discussion

The present study aimed to evaluate the presence of ADHD specific symptoms in a group of patients diagnosed with SUDs, hospitalized in an addiction ward within a clinical psychiatry hospital. Subsequently, after identifying the patients who met the diagnostic criteria for ADHD, the group was divided into two subgroups in order to compare the profile of substance use and comorbidities according to the presence or absence of ADHD symptoms. The secondary objective of the study was to characterize and compare the two subgroups in terms of personality traits and underlying personality disorders.

The lower level of self-control reported by adults with ADHD compared to those without ADHD is explained by the clinical picture and the specific symptomatology of this disorder. Thus, the impulsiveness characteristic of ADHD in the adult is manifested by low tolerance to frustration, irritability, hurried responses, difficulties in waiting for their turn, easily entering into conflicts, impulsive expenses [23]. Previous research linked impulsive traits to a certain proneness to addictive behaviours [24,25]. One recent study on young Italian adults for example, showed a link between high levels of impulsivity and Problematic Use of the Internet (PUI) [26]. Moreover, expression of higher impulsivity levels in patients with personality disorders has been associated with increased substance-associated risky behaviours [27,28].

This self-regulating deficit is reflected in the scores obtained in the personality traits questionnaire, Self-control section.

The lower scores obtained by the subjects in the group with ADHD for the Relationship skills items compared to the group without ADHD, is explained by the difficulties of people with this disorder to establish and maintain relationships with the people around them. So, both in the case of children with ADHD and in adults, patients have difficulties maintaining relationships because of impulsivity and inadequate reactions, and because they lose interest and become bored easily. Moreover, these people tend to change environments, jobs, relationships out of impulsivity [23].

Previous research investigating the relationship between identity development and substance abuse, suggested that individuals with a non-integrated identity status are more likely to experiment drugs or alcohol at a younger age, to stay abstinent for a shorter time period, and to register a low compliance to the therapeutic process [29,30]. In our research the links between Identity Integration and substance abuse, and the presence or absence of ADHD symptoms were not demonstrated.

Even though one of the characteristics of addictions is the inability to take responsibility, in our study there was no significant statistical difference between the two groups regarding the scores obtained for Responsibility sections. These findings may suggest that in our group the pathological personality traits can be integrated in a Cluster B Personality Disorder, more specific in a narcissist or antisocial type. The clinical characteristics suggestive for Borderline Personality Disorder (identity disturbances, emotional lability) were not statistically significant in our group. Considering this results in comparison with previous studies, we concluded that there is still controversy over the hypothesis that ADHD is a precursor to the development of Borderline Personality Disorder, as there are authors who consider that these two disorders are so frequently associated because they overlap with the same genetic and environmental aetiology [31].One study form 2014 showed that from a group of patients with personality disorders who responded poorly to treatment, 6% of them had undiagnosed ADHD symptoms [32]. These data have been replicated by other research, showing that severe personality disorders often associate comorbid ADHD, and the presence of this pathology predisposes to the development of severe disruptive behaviours, and increased degrees of impulsivity [33].

Our findings are consistent with other previous results, which highlight the presence of Antisocial Personality Disorder in patients with ADHD in young adulthood [34,35]. We can conclude that the persistence of ADHD symptoms in adulthood can be considered a predictive factor for maladaptive cluster B personality disorders, especially within the antisocial spectrum. Our research brings novelty by examining this trend in patients that have both ADHD and Substance Abuse Disorder diagnoses.

Severe personality disorders frequently associate comorbid ADHD, and the presence of this pathology predisposes to the development of accentuated disruptive behaviours, and an increased degree of impulsivity. According to data from precedent studies, the development of disharmonious personality characteristics in children overlapped with a history of oppositional defiant and/or aggressive behaviour. Moreover, an overlap with the three criteria (“mood swings”, “inappropriate anger” or Impulsivity) of antisocial personality was identified in 90% of patients with ADHD. The combined type or the hyperactivity/impulsivity type increases the risk of developing Antisocial Personality Disorder [36].

## 5. Clinical Implications

Most studies reported in the literature on the association of substance use with ADHD pathology in adults, were performed on groups of patients diagnosed with ADHD to investigate the presence of behavioural disorders secondary to psychostimulant use. The present research brings as a novelty the retrospective investigation of ADHD symptoms in adulthood in a group of patients admitted to a psychiatric service with an addiction profile, underlining the importance of raising awareness regarding the identification and accurate diagnosis of ADHD in adult psychiatric services. However, thorough analysis of each case is needed, with investigation o the presence of ADHD symptomatology during childhood, as symptoms declared by adult patients, which can be superimposed over the ADHD criteria, may be part of the clinical picture of another psychiatric disorder, such as bipolar affective disorder or personality disorder, which have late onset, adolescence or adulthood. The diagnosis of ADHD has criteria applicable to adults only with DSM-5 and is still an underdiagnosed and untreated disorder, which further increases the risk of chronicity. All of the above results may increase awareness of the use of the diagnosis of ADHD in adult psychiatric services. Early diagnosis of ADHD and comorbidities could better guide the intervention (both pharmacological and psychotherapeutic) in these patients in order to improve their quality of life.

## 6. Conclusions

The clinical picture in the adult with ADHD and Substance Abuse Disorder is very complex and of great symptomatic diversity. The more psychiatric comorbidities there are, the harder they are to treat and more difficult it is to achieve remission despite proper interventions. In the case of patients with antisocial personality traits that are difficult to treat it is important to consider them during evaluation, as symptoms of an undiagnosed ADHD, because it can provide more perspectives on intervention.

The hypothesis that ADHD is a precursor to the development of Antisocial Personality Disorder is becoming increasingly valid. The developmental trajectory and the factors that modulate the course of this comorbidity throughout life are very little understood and require further study.

## Figures and Tables

**Figure 1 medicina-58-00962-f001:**
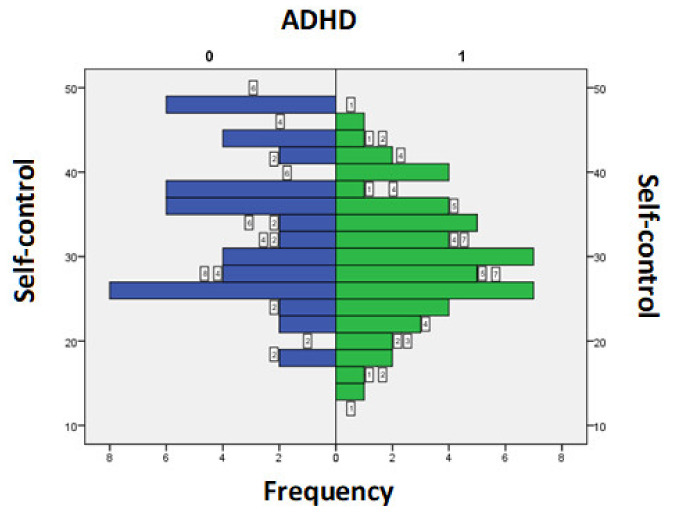
Score for Self-control section according to the presence/absence of ADHD symptoms.

**Figure 2 medicina-58-00962-f002:**
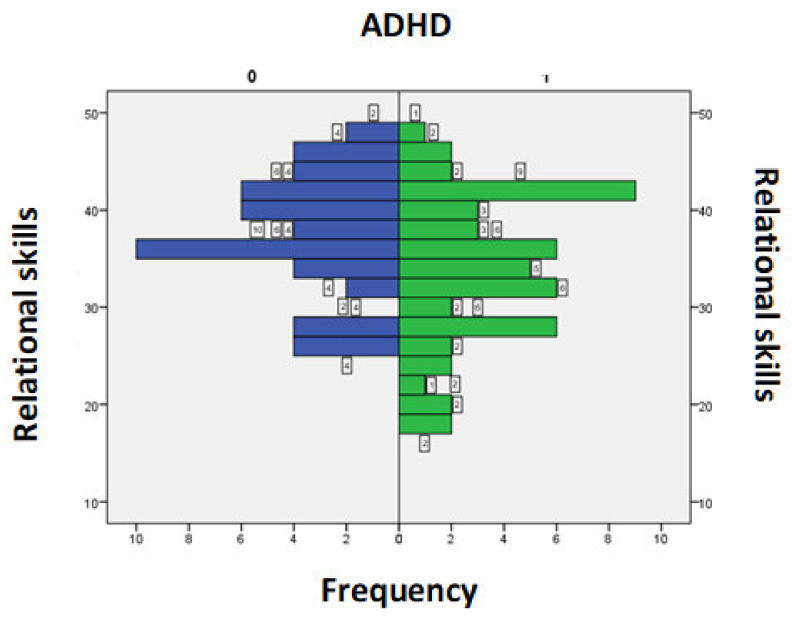
Score for the Relational Skills section according to the presence/absence of ADHD symptoms.

**Figure 3 medicina-58-00962-f003:**
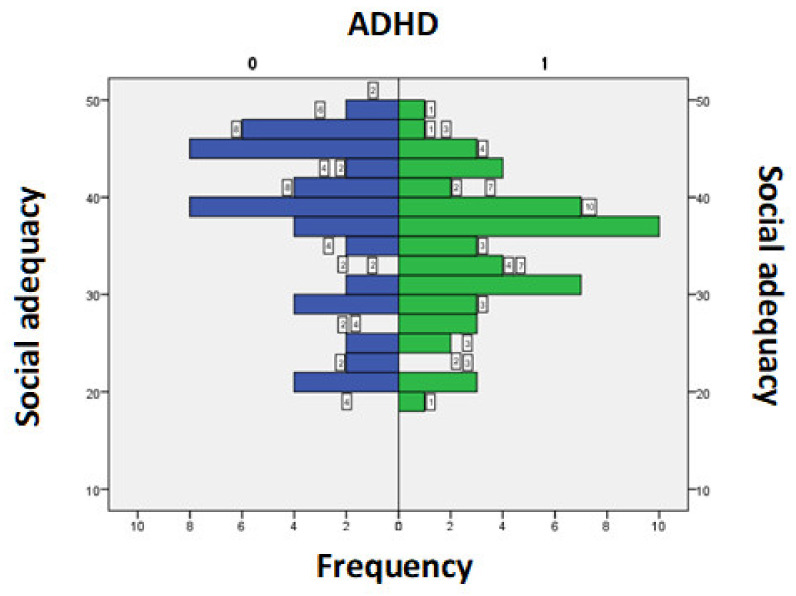
Score for the field of Social Adequacy according to the presence/absence of ADHD symptoms.

**Table 1 medicina-58-00962-t001:** Pearson correlation matrix for domains investigated through the personality traits questionnaire.

	Identity Integration	Responsibility	Relational Skills	Social Adequacy
Self-control	0.66 0.00	0.75 0.00	0.52 0.00	0.74 0.00
Identity Integration		0.69 0.00	0.73 0.00	0.75 0.00
Responsibility			0.55 0.00	0.66 0.00
Relational Skills				0.67 0.00

## Data Availability

Not applicable.

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
