# Peer review of "Maladaptive Personality Traits in a Group of Patients with Substance Use Disorder and ADHD"

_medicina, 2022, doi:10.3390/medicina58070962_

Round 1

Reviewer 1 Report

The study illustrates some possible aspects of the comorbidity of personality disorders and the diagnosis of ADHD in patients with substance use disorders.

The main strength of this paper is that the correlation between these three elements (ADHD, personality disorder, substance use disorder) has never been studied adequately, although the reasonableness of the hypothesis at the basis of the investigation appears very evident, as demonstrated also by the data collected.

A worthy limitation is represented by some methodology issues.

Introduction

Please do not use the expression “drug use/abuse”; use instead “substance use/abuse”.

Substance use disorders should be shortened as SUDs.

No aims are stated at the end of the introduction; please insert a paragraph when you clearly state the aims of your research.

Matherials and Methods

This section needs an ample revision as it is very poor.

Why did you selected that age range? You should clarify this element.

No information about psychometric properties of the instruments used are provided (validated in Romanian? How are they scored? Chronbach’s alpha?...)

Was the study approved by any institutional ethical committee?

It would be better to divide the section into sub-sections: participants, procedures, instruments, statistical analysis.

Other limitations:

The small sample and also his provenance that is not optimal to have a paradigmatic representativeness: only one psychiatric ward.

Another limitation is definitely not having relied on a thorough clinical evaluation and not having specified or motivated the choice of psychometric scales for the diagnosis of ADHD and personality disorders (for example for the latter there are MMPI-2: Minnesota Multiphasic Personality Inventory (Fiore, 2012); Il MCMI-IV: Millon Clinical Multiaxial Inventory (Terenzi, 2017); SCID-5-PD (APA, 2013).  Certainly the use of a reduced scale such as the Severity Indices of Personality Problems (SIPP-SF) did not help too much to the clinical classification of the subjects participating in the study.

Results

Considering what has been highlighted for the "Materials and Methods" section, this part appears to be very narrow although sufficiently adequate and clear in reporting the findings; it would probably benefit if described and discussed more amply, also to better connect and integrate with the rest of the article and above all to ensure a good continuity of understanding for the reader.

Discussions

You should start the section with a synthesis of your aims in conducting this research and of the main results obtained, that you are going to discuss later in the section.

The authors could eventually add some further reflections on what has been elaborated.

Conclusion

This part seems too dull,it could probably be rendered more attractively.

Figures

Figures are quite poor in their quality. Please translate the parts left in Romanian into English.

Bibliography does not cover adequately the main papers of the field (25 entries only, that should be increased to sustain introduction and discussion).

Authors may want to extend their reflection from substance addiction to addictive behaviors in general. A papers relating to impulsivity among problematic internet users:

https://pubmed.ncbi.nlm.nih.gov/33581451/

The fundamental relation between substance use and psychoses should also be accounted in the introduction;

Author may consider a report about dissociative symptoms in substance users

https://pubmed.ncbi.nlm.nih.gov/34688166/

and one about duration of untreated schizophrenia among substance users

https://pubmed.ncbi.nlm.nih.gov/34886357/

I thank editor for the opportunity to review and wait for the revised version.

Author Response

Response to Reviewer 1 Comments

Dear Reviewers,

We would like to express our grattiude and appreciation for your time and effort in writing your comments on the manuscript. We trust that all of your comments have been addressed accordingly in this response letter and the revised manuscript.

In the following, we give a point-by-point reply to your comments:

The study illustrates some possible aspects of the comorbidity of personality disorders and the diagnosis of ADHD in patients with substance use disorders. The main strength of this paper is that the correlation between these three elements (ADHD, personality disorder, substance use disorder) has never been studied adequately, although the reasonableness of the hypothesis at the basis of the investigation appears very evident, as demonstrated also by the data collected. A worthy limitation is represented by some methodology issues.

Point 1: Introduction

Please do not use the expression “drug use/abuse”; use instead “substance use/abuse”.

Substance use disorders should be shortened as SUDs.

No aims are stated at the end of the introduction; please insert a paragraph when you clearly state the aims of your research.

Response 1: We would like to express our gratitude to the reviewer for the reccomendations. The term “drug use/abuse” was substituted with “substance use/abuse” throught the text, as it was done with Substance use disorders which was shirtened as SUDs.

We appreciated the reviewer’s suggestion and thus we added the aims of the research a the and of the introduction (lines 86-89 of the revised manuscript): The main objective of this study is to determine whether there is a significant re-lationship between personality disorders and drug use / ADHD. The secondary objective was to establish the most common type of personality disorder involved in this relationship.

Point 2: Matherials and Methods

This section needs an ample revision as it is very poor.

Point 2.1: Why did you selected that age range? You should clarify this element.

Response 2.1: The age of the adult patients included in the study is between 18-25 years. 75% of the subjects were between 18 and 22 years old, with 21 years old being the age with most representatives. This age range was selected as the ward within which we conducted our research is dedicated to younger adults. Moreover, the majority of those who struggle with these disorders and who are referred to the aforementioned addictions clinic are aged 18-25 and they were generally more inclined to cooperate and consent to the inclusion in the study.

Point 2.2: No information about psychometric properties of the instruments used are provided (validated in Romanian? How are they scored? Chronbach’s alpha?...)

Response 2.2: The instruments are validated in Romanian. This interview is based on the DSM criteria for ADHD and evaluates 18 criteria for current and retrospective (from childhood) behaviors. Deficiencies in the areas of functioning of daily glife are also assessed. Testing is recommended in the presence of a family member to simultaneously evaluate heteroanamnestic information. The 3-part questionnaire assesses attention deficit, hyperactivity / impulsivity and age of onset / dysfunction associated with symptoms. Responses are adjusted to recent time (presence of symptom in the last 6 months) and retrospectively (presence of symptom between 5 and 12 years). For each symptom, its presence or absence is established in both stages of life. For scoring, it is established whether for each of the 3 parts at least 6 criteria have been registered. It is also recorded if there is evidence of continuity of the symptom in life, if it is associated with dysfunctions, if the dysfunctions are present in at least 2 areas of life and if the symptoms could be better explained by another psychiatric entity.

As the instruments are used internationally and have been validated, we considered that Chronbach’s alpha was not necessary for these particular instruments.

Point 2.3: Was the study approved by any institutional ethical committee?

Response 2.3: Thank you for pointing out this oversight.

The selection protocol of the subjects included in the study and research methodology were applied with the agreement of the Ethics Committee within "Carol Davila" University of Medicine and Pharmacy and “Prof. Dr. Alexandru Obregia” Clinical Psychiatry Hospital (lines 105-108 of the revised manuscript).

Point 2.4: It would be better to divide the section into sub-sections: participants, procedures, instruments, statistical analysis.

Response 2.4: Thank you for this suggestion. We edited the manuscript accordingly (lines 94, 114, and 130 of the revised manuscript).

Point 3: Other limitations:

Point 3.1: The small sample and also his provenance that is not optimal to have a paradigmatic representativeness: only one psychiatric ward.

Response 3.1: The patients were selected from only one psychiatric ward as the “Prof. Dr. Alexandru Obregia” Clinical Psychiatry Hospital is a general psychiatry hospital, adressing the whole spectrum of mental health disorders, SUDs included. Thus, we selected the patients from the young adults addictions ward, as they were more inclined to cooperate and consent to the inclusion in the study.

Point 3.2: Another limitation is definitely not having relied on a thorough clinical evaluation and not having specified or motivated the choice of psychometric scales for the diagnosis of ADHD and personality disorders (for example for the latter there are MMPI-2: Minnesota Multiphasic Personality Inventory (Fiore, 2012); Il MCMI-IV: Millon Clinical Multiaxial Inventory (Terenzi, 2017); SCID-5-PD (APA, 2013).  Certainly the use of a reduced scale such as the Severity Indices of Personality Problems (SIPP-SF) did not help too much to the clinical classification of the subjects participating in the study.

Response 3.2: We appreciate the reviewer’s suggestion, and, in the revised manuscript, we have provided additional information regarding the diagnosis procedure (lines 114-117 in the revised manuscript): All the participants underwent a comprehensive psychiatric assessment performed by a psychiatrist, consisting of the medical and developmental history, the educational history, the mental status examination, a Diagnostic and Statistical Manual of Mental Disorders, Fifth Edition (DSM-5) diagnosis and the treatment history.

The reasons behind the choice of the psychometric scales we used lie in their accesibility, Romanian validation, and ease-of-use with the subjects. 

Point 4: Results

Considering what has been highlighted for the "Materials and Methods" section, this part appears to be very narrow although sufficiently adequate and clear in reporting the findings; it would probably benefit if described and discussed more amply, also to better connect and integrate with the rest of the article and above all to ensure a good continuity of understanding for the reader.

Response 4: We thank the reviewer for the recommendation. Our aim was to present the results of our research in a clear and concise matter, without cluttering them with supplemental information. In the hope that in spite of considering this section narrow, the reviewer still appreciated the clarity of the data, we performed minor edits on this section.

Point 5: Discussions

You should start the section with a synthesis of your aims in conducting this research and of the main results obtained, that you are going to discuss later in the section.

The authors could eventually add some further reflections on what has been elaborated.

Response 5: This section was reformulated according to the reviwer’s suggestions.

Point 6: Conclusion

This part seems too dull,it could probably be rendered more attractively.

Response 6: We appreciate the reviewer’s comment, and, therefore, in the revised manuscript, the conclusions section was edited.  

Point 7: Figures

Figures are quite poor in their quality. Please translate the parts left in Romanian into English.

Response 7: The parts left in Romanian were translated into English.

Point 8: Bibliography does not cover adequately the main papers of the field (25 entries only, that should be increased to sustain introduction and discussion).

Authors may want to extend their reflection from substance addiction to addictive behaviors in general. A papers relating to impulsivity among problematic internet users:

https://pubmed.ncbi.nlm.nih.gov/33581451/

The fundamental relation between substance use and psychoses should also be accounted in the introduction;

Author may consider a report about dissociative symptoms in substance users

https://pubmed.ncbi.nlm.nih.gov/34688166/

and one about duration of untreated schizophrenia among substance users

https://pubmed.ncbi.nlm.nih.gov/34886357/

Response 8: We appreciate the reviewer’s input and thus, in the revised manuscript, we extended our bibliography, referencing the suggested articles, as well as a few additional ones.

Reviewer 2 Report

This is a very interesting research about patients with SUD and comorbid ADHD. However, the background lacks of explanations about personality characteristics of patients with dual disorders. Even this paper is about a very specific comorbidity (SUD+ADHD), there are some previous papers that present personality trait is these samples; this studies are not cited nor mentioned at all. Dual diagnosis and personality has shown to have different clinical correlates that are not mentioned here either. 

Such lack of background is also presented later at the discussion section, where the results need to be more linked to previous findings. The clinical implications derived from the researched carried out in the present manuscript must be highlighted. 

Author Response

Response to Reviewer 2 Comments

Point 1:  This is a very interesting research about patients with SUD and comorbid ADHD. However, the background lacks of explanations about personality characteristics of patients with dual disorders. Even this paper is about a very specific comorbidity (SUD+ADHD), there are some previous papers that present personality trait is these samples; this studies are not cited nor mentioned at all. Dual diagnosis and personality has shown to have different clinical correlates that are not mentioned here either.

Such lack of background is also presented later at the discussion section, where the results need to be more linked to previous findings. The clinical implications derived from the researched carried out in the present manuscript must be highlighted.

Response 1:

Certain personality features are considered, by some authors, the underlying mecha-nism in patients with dual diagnosis of psychotic disorders and SUD. In a 2021 review by Oh et al., after a meta-analysis on personality traits of patients with psychotic dis-orders and SUD comorbidity (dual diagnosis - DD), the authors found that patients with DD presented “elevated negative urgency”, “low premeditation”, as well as “elevated unconscientious disinhibition” [17].

Considering this results in comparison with previous studies, we concluded that there is still controversy over the hypothesis that ADHD is a precursor to the development of Borderline Personality Disorder, as there are authors who consider that these two disorders are so frequently associated because they overlap with the same genetic and environmental etiology [24]. One study form 2014 showed that from a group of patients with personality disorders who responded poorly to treatment, 6% of them had undiagnosed ADHD symptoms [25]. These data have been replicated by other research, showing that severe personality disorders often associate comorbid ADHD, and the presence of this pathology predisposes to the development of severe disruptive behaviors, and increased degrees of impulsivity [26].

  1. Clinical implications

Most studies reported in the literature on the association of substance use with ADHD pathology in adults, were performed on groups of patients diagnosed with ADHD to investigate the presence of behavioral disorders secondary to psychostimulant use. The present research brings as a novelty the retrospective investigation of ADHD symptoms in adulthood in a group of patients admitted to a psychiatric service with an addiction profile, underlining the importance of raising awareness regarding the identification and accurate diagnosis of ADHD in adult psychiatric services. However, thorough analysis of each case is needed, with investigation o the presence of ADHD symptomatology during childhood, as symptoms declared by adult patients, which can be superimposed over the ADHD criteria, may be part of the clinical picture of another psychiatric disorder, such as bipolar affective disorder or personality disorder, which have late onset, adolescence or adulthood. The diagnosis of ADHD has criteria applicable to adults only with DSM-5 and is still an underdiagnosed and untreated disorder, which further increases the risk of chronicity. All of the above results may increase awareness of the use of the diagnosis of ADHD in adult psychiatric services. Early diagnosis of ADHD and comorbidities could better guide the intervention (both pharmacological and psychotherapeutic) in these patients in order to improve their quality of life.

Round 2

Reviewer 1 Report

Comments have been addressed